
# Linearization of the effect of slit function changes for improving OMI ozone profile retrievals

**Juseon Bak[a,*], Xiong Liu[a], Kang Sun[b], Kelly Chance[a], Jae-Hwan Kim [c]**

[a]*Harvard-Smithsonian Center for Astrophysics, Cambridge, MA, USA*

[b]*Research and Education in eNergy, Environment and Water Institute, University at Buffalo, Buffalo, NY, USA*

[c]*Atmospheric Science Department, Pusan National University, Busan, Korea*

*Corresponding Author (juseon.bak@cfa.harvard.edu)

## Abstract

We introduce a method that reduces the spectral fit residuals caused by the slit function errors in an optimal estimation based spectral fitting process to improve ozone profile retrievals from the Ozone Monitoring Instrument (OMI) ultraviolet measurements (270-330 nm). Previously, a slit function was parameterized as a standard Gaussian by fitting the Full Width at Half Maximum (FWHM) of the slit function from climatological OMI solar irradiances. This cannot account for the temporal variation of slit function in irradiance, the intra-orbit slit function changes due to thermally-induced change and scene inhomogeneity, and potential differences in the slit functions of irradiance and radiance measurements. As a result, radiance simulation errors may be induced due to using the convolved reference spectra with incorrect slit functions. To better represent the shape of the slit functions, we implement a more generic super Gaussian slit function with two free parameters (slit width and shape factor); it becomes standard Gaussian when the shape factor is fixed to be 2. The effects of errors in slit function parameters on radiance spectra, referred as "Pseudo Absorbers (PAs)", are linearized by convolving high-resolution cross sections or simulated radiances with the partial derivatives of the slit function with respect to the slit parameters. The PAs are included in the spectral fitting scaled by fitting coefficients that are iteratively adjusted as elements of the state vector along with ozone and other fitting parameters. The fit coefficients vary with cross-track and along-track pixels and show sensitivity to heterogeneous scenes. The total PA spectrum is quite similar in the Hartley band below 310 nm for both standard and super Gaussians, but is more distinctly structured in the Huggins band above 310 nm with the use of super Gaussian slit functions. Finally, we demonstrate that some spikes of fitting residuals are slightly smoothed by accounting for the slit function



errors. Comparisons with ozonesondes demonstrate substantial improvements with the use of PAs for both
standard and super Gaussians, especially for reducing the systematic biases in the tropics and mid-latitudes
and reducing the standard deviations at high-latitudes. Including PAs also makes the retrievals consistent
between standard and super Gaussians. This study corroborates the slit function differences between
radiance and irradiance demonstrating that it is important to account for such differences in the ozone profile
retrievals.

## 1. Introduction

The fitting of the measured spectrum to the simulated spectrum is the most basic concept for the analysis
of the Earth's atmospheric constituents from satellite measurements. Therefore, the accurate calibration and
simulation of measurements are essential for the successful retrieval of atmospheric constituents. The
knowledge of the instrumental spectral response function (ISRF) or slit function could affect the accuracies
of both calibration and simulation, as it is required for the convolution of a high-resolution reference
spectrum onto instrument's spectral resolution in the wavelength calibration and for the convolution of
high-resolution absorption cross section spectra or simulated radiance spectrum in the calculation of
radiance at instrumental resolution. Compared to other trace gases, the retrieval of ozone profiles could be
more susceptible to the accuracy of ISRFs due to the large spectral range, where the radiance spans a few
orders of magnitude and to the fact that the spectral fingerprint for the tropospheric ozone is primarily
provided by narrow and weak absorption features of the temperature-dependent Huggins bands (320-360
nm). Therefore, the efforts of characterizing and verifying the ISRFs have preceded the analyses of ozone
profiles from the satellite/aircraft measurements (Liu et al., 2005, 2010; Cai et al., 2012; Liu et al., 2015;
Sun et al. 2017; Bak et al., 2017).
For space-borne instruments, ISRFs are typically characterized as a function of the detector dimensions
using a tunable laser source prior to the launch (Dirksen et al., 2006; Liu et al., 2015; van Hees et al., 2018).
However, the preflight measured ISRFs could be inconsistent with those after launch due to the orbital
movement and the instrument temperature change (Beirle et al., 2017; Sun et al., 2017). Therefore, the post-
launch ISRFs have been typically parameterized through a cross-correlation of the measured solar
irradiance to a high-resolution solar spectrum (Caspar and Chance, 1997), assuming Gaussian-like shapes.
The direct retrieval of the ISRFs from radiances has not typically been done due to the complication of
taking the atmospheric trace gas absorption and Ring effect into account in the cross-correlation procedure
and the slow-down of the fitting process. However, slit function differences between radiance and
irradiance could exist due to scene heterogeneity, differences in stray light between radiance and radiance,





intra-orbit instrumental changes, and the instrument temperature change (Beirle et al., 2017; Sun et al.,
2017). In addition, using temporally invariant slit functions derived from climatological solar spectra in the
retrievals could cause the long-term trend errors if instrument degradation occurs. Therefore, there is room
for improving our trace gas retrievals by accounting for the effects of the different ISRFs between radiance
and irradiance on the spectral fitting and on the pixel-to-pixel basis. The "Pseudo Absorber (PA)" is a
common concept in spectral fitting to account for the effect of the physical phenomena that is difficult or
computationally demanding to be simulated in the radiative transfer calculation, like spectral misalignments
(shift and stretch) between radiance and irradiance, Ring effect, spectral undersampling, and additive stray-
light offsets. The pseudo absorption spectrum can be derived from a finite-different scheme (e.g. Azam and
Richter, 2015) or a linearization scheme via a Taylor expansion (e.g. Beirle et al., 2013; 2017); the latter
approach is more efficient than the former one, but less accurate because only the first term of the Taylor
series is typically taken into account for simplicity. Beirle et al. (2013) introduced a linearization scheme
to account for spectral misalignments between radiance and irradiance and then included them as a pseudo-
absorber in DOAS-based $NO_2$ and BrO fittings. Similarly, Beirle et al. (2017) linearized the effect of the
change of the ISRF parameterized as a super Gaussian on GOME-2 solar irradiance spectra to characterize
the slit function change over time and wavelength. Sun et al. (2017) derived on-orbit slit functions from
solar irradiance spectra measured by the Ozone Monitoring Instrument (OMI) (Levelt et al., 2006) assuming
standard Gaussian, super Gaussian, and preflight ISRFs with adjusted widths. The derived on-orbit slit
functions, showing significant cross-track dependence that cannot be represented by preflight ISRFs,
substantially improve the retrievals by the Smithsonian Astrophysical Observatory (SAO) ozone profile
algorithm. However, it is not fully understood why the use of super Gaussian or stretched preflight functions,
which are supposed to better model the OMI spectra as indicated by smaller mean fitting residuals, does
not improve the retrievals over the use of standard Gaussian especially in the standard deviations of the
differences with relative to ozonesonde observations. This study suggested that the slit functions derived
from solar spectra might not fully represent those in radiance spectra.

As such, the objective of this paper is to implement the slit function linearization proposed by Beirle

et al. (2017) into the optimal estimation based spectral fitting of the SAO ozone profile algorithm. We
further improve the slit function parameterization by accounting for the differences between radiance and
irradiance slit functions on a pixel-by-pixel basis, and ultimately to improve OMI ozone profile retrievals.
This paper is organized as follows: after a mathematical description of the linearization of slit function
changes using the generic super Gaussian function, we introduce how to apply them practically in an
optimal estimation based spectral fit procedure (Section 2). This linearization scheme is differently
implemented, depending on the simulation scheme of measured spectra using high resolution radiances or
effective cross section data, respectively. Section 3 characterizes the derived pseudo absorber spectra, along





with the evaluations of ozone profile retrievals using independent ozonesonde observations as a reference
dataset. Finally, the summary of this study is given in Section 4.
**2.  Method**
**2.1 Super Gaussian linearization**
The slit function parameterization and linearization are briefly summarized from Beirle et al. (2017),
focusing on what we need to derive the pseudo absorbers in the terms of the optimal estimation based fitting
process. The slit function can be parameterized with the slit width $w$, and shape factor $k$ assuming the
supper Gaussian, S as:
$$S(\Delta\lambda) = A\,(w,k)\,\times\,exp\left[-\left|\frac{\Delta\lambda}{w}\right|^{k}\right], \qquad (1)$$

where $A(w,k)$ is $\frac{k}{2\sigma_{g}\Gamma\left(\frac{1}{w}\right)}$ with $\Gamma$ representing the gamma function. This equation allows many forms of
distributions by varying $k$: the top-peaked function ($k<2$), the standard Gaussian function ($k=2$), and the
flat-topped function ($k>2$). $w$ is converted to the Full Width at Half Maximum (FWHM) via the relationship
of FWHM $= 2\sqrt[k]{ln2}\,w$ . We investigate the impact of including one more slit parameter $k$ on the OMI ISRF
fit results over the standard Gaussian using OMI daily solar measurements. As an example, time-series
(2005-2015) of the fitted slit width and shape factor in 310-330 nm are displayed in Figure 1.a. The FWHM
and shape factor of the super Gaussian function is on average 0.44 nm and 2.9, respectively, while the
FWHM of the standard Gaussian is 0.395 nm. The degradation of the OMI slit functions became relatively
visible after 2011. The high wavelength stability (0.003 nm) is seen in Figure 1b, verifying that better
calibration stability is performed with super Gaussian slit functions as abnormal deviations of wavelength
shifts are derived with standard Gaussian slit functions.
The effect of changing the slit parameters $p$ on the slit function can be linearized by the first-order
Taylor expansion approximation around $S_{o} = S(p_{o})$:
$$\triangle S = S - S_{o} \approx \triangle p\,\frac{\partial S}{\partial p}, \quad (2)$$

and thus the effect of changes of S on the convolved high-resolution spectrum can be parameterized as
$$\triangle I = I - I_{o} = S \otimes I_{h} - S_{o} \otimes I_{h} = \triangle S \otimes I_{h}\,, (3)$$

where the convolved spectrum is $I = S \otimes I_{h}$. Consequently, the partial derivatives of I with respect to slit
parameters, $p$ are defined as



$$\frac{\partial I}{\partial p} = \frac{\partial S}{\partial p} \otimes I_h \ . \ (4)$$

In Beierle et al. (2017), $\frac{\partial I}{\partial p}$ refers to $J_p$ as "resolution correction spectra (RCS)". In Figure 2, we present
an example of $J_p$ over the typical ozone profile fit range (270-330 nm) through the convolution of high-
resolution ozone cross sections ($\delta_h$) with the derivatives of the super Gaussian ($\frac{\partial S}{\partial p}$). The baseline $S_o$ is
defined with $w$=0.26 nm and $k$=2.6, which are averaged parameters from climatological OMI solar
irradiance spectra in the UV2 band (310-330 nm). Note that this $w$ value corresponds to a FWHM of 0.45
nm. The change of the assumed OMI slit function causes a highly structured spectral response over the
whole fitting window. However, the relative magnitude of the responses with respect to both slit parameters
is more distinct in the Huggins band (>310 nm) where narrow absorption features are observed as shown
in Figure 2.a.  An anti-correlation (-0.92) is found between $\frac{\partial \ln \delta}{\partial w}$ and $\frac{\partial \ln \delta}{\partial k}$ while the response of the unit
change of the slit width to the convolved spectrum is dominant against that of the shape factor.

**2.2 Implementation of the slit function linearization in the SAO ozone profile algorithm**

We implement the slit function linearization in the SAO ozone profile algorithm (Liu et al. 2010), which
is      routinely      being      performed      to      produce      the      OMI      PROFOZ      product
(https://avdc.gsfc.nasa.gov/index.php?site=1389025893&id=74). Two spectral windows (i.e., 270-309 nm
in the UV1 band and 312-330 nm in the UV2 band) are employed to retrieve ozone profiles from OMI BUV
measurements. To match the different spatial resolutions between UV1 and UV2 bands, every two cross-
track pixels are averaged for UV2 band, resulting into 30 positions with the spatial resolution of 48 km
(across-track) × 13 km (along-track) at nadir position. The non-linear optimal estimation based fitting is
iterated toward minimizing the fitting residuals between measured and simulated radiances and
simultaneously between a priori and estimated ozone values. A priori ozone information is taken from a
tropopause-based (TB) ozone profile climatology (Bak et al., 2013). The Vector Linearized Discrete
Ordinate Radiative Transfer model (VLIDORT) (Spurr, 2008) is used to simulate the radiances and their
derivatives with respect to geophysical parameters. The radiance calculation is made for the Rayleigh
atmosphere, where the incoming sunlight is simply absorbed by ozone and other trace gases, scattered by
air molecules, and reflected by surfaces/clouds assumed as a Lambertian surface. Besides these physical
phenomena, the others are treated as PAs to the spectral response such as Ring effect, additive offset, and
spectral shifts due to misalignments of radiance relative to irradiance and ozone cross sections. In the SAO
algorithm, these PAs are derived using the finite differences of the radiances with and without perturbation



to a phenomenon, except for the Ring spectrum that is calculated using a first-order single scattering
rotational Raman scattering model (Sioris and Evans, 2000).

In this paper, we introduce new PAs to account for the radiance simulation errors caused by the slit

function errors. The OMI ISRFs have been parameterized as a standard Gaussian from climatological OMI
solar irradiances for each UV1 and UV2 band and thereby these PAs could take into account the spectral
fitting responses caused by temporal variations of the slit function. This ozone fitting procedure uses ISRFs
to convolve high resolution absorption spectra, taken from Brion et al. (1993) for ozone absorption cross
section and Wilmouth et al. (1999) for BrO absorption cross section. Our algorithm has implemented two
different convolution processes, i.e. the effective cross section approach in Liu et al (2010) and the high-
resolution convolution approach described in Kim et al. (2013), respectively and thereby this paper also
introduces how to derive the derivatives of the OMI radiances with respect to ISRF changes in these two
approaches. Although the latter is the current approach, we also implement and present the linearization
with the first approach, which is typically used for other trace gas retrieval algorithms.

In Liu et al (2010), VLIDORT simulates the radiances at OMI spectral grids ($\lambda_{omi}$) using effective cross

sections that are produced by convolving high-resolution cross sections with the OMI ISRF. Therefore, we
apply a similar convolution process of matching the high-resolution cross section spectrum with OMI
spectrum to derive the partial derivative of $\sigma_x$ with respect to slit parameter, p as follows:
$$\frac{\partial \sigma_x}{\partial p} = \frac{\partial S}{\partial p} \otimes \sigma_{x,h} \text{ , (5)}$$

where $\sigma_{x,h}$ is a high-resolution absorption spectrum for ozone and BrO, respectively. Due to the dominant
ozone absorption over the BrO absorption, the derivative of BrO cross section with respect to p is neglected
here. This partial derivative of ozone is then converted to the partial derivative of radiance through the chain
rule with the analytical ozone weighting function ($\frac{d\ln I}{dO_3}$), calculated from VLIDORT, as follows:

$$\frac{\partial \ln I}{\partial p} = \frac{\partial \ln I}{\partial O_3} \frac{\partial \sigma}{\partial p} \frac{O_3}{\sigma} \text{ . (6)}$$

This simulation process is hereafter referred to as "effective resolution cross section (ER) simulation".

As described in Kim et al. (2013), the radiative transfer calculation in the SAO ozone profile algorithm

has been performed using high-resolution extinction spectra at the optimized sampling intervals for
resolving the ozone absorption features, which are a 1.0 nm below 300 nm and 0.4 nm above 300 nm. These
sampling intervals are coarser than actual OMI sampling grids with approximately half the number of
wavelengths. The coarser sampled simulated radiances are then interpolated to a fine grid of 0.05 nm
assisted by the weighting functions with respect to absorption and Rayleigh optical depth:





$$I(\lambda_h) = I(\lambda_c) + \frac{\partial I\,(\lambda_c)}{\partial \Delta_l^{gas}}\left(\Delta_l^{gas}(\lambda_h) - \Delta_l^{gas}(\lambda_c)\right) + \frac{\partial I\,(\lambda_c)}{\partial \Delta_l^{ray}}\left(\Delta_l^{ray}(\lambda_h) - \Delta_l^{ray}(\lambda_c)\right), (7)$$
where $\Delta_l^{gas}$ and $\Delta_l^{ray}$ are the optical thickness (the product of cross section and layer column density) at
each layer for trace gas absorption and Rayleigh scattering, respectively. The convolution is then applied
to these simulated high-resolution radiances, $I(\lambda_h)$ with assumed slit functions and derivatives, respectively,
and thereby $I(\lambda_{omi})$ and $\frac{\partial \ln I}{\partial p}$ is calculated. This simulation process is hereafter referred to as "high-
resolution cross section (HR) simulation". The ER simulation is more commonly implemented in the trace
gas retrievals in the UV and visible, but the HR simulation allows for more accurate fitting residuals to,
better than 0.1 % (Kim et al., 2013) as well as shorter computation time. $\frac{\partial \ln I}{\partial p}$ is scaled by the fitting
coefficients, $\Delta p$, to account for the actual size of the spectral structures caused by the slit function
differences between radiance and irradiance spectra. The total "pseudo absorber (PA)" for the Super
Gaussian slit function linearization is expressed as:
$$PA = \partial \ln I = \frac{\partial \ln I}{\partial k}\,\Delta k + \frac{\partial \ln I}{\partial w}\,\Delta w. (8)$$
In the form of the logarithm of normalized radiances, PA is physically related to the optical depth change
$\Delta \tau$. Figure 3 compares the partial derivatives of radiances to slit parameters in HR and ER simulations.
Little difference is found even though convolution error for ozone cross sections is only accounted for in
the ER simulation due to the overwhelming impact of ozone cross section convolution errors over other
cross section data.
Furthermore, this linearization process can be formulated with n-order polynomial fitting parameters
($\Delta p_i$) to account for the wavelength-dependent change of the slit parameters around a central wavelength $\bar{\lambda}$
and consequently, the total PA is expressed as
$$PA = \frac{\partial \ln I}{\partial k}\sum_{i=1}^{n}\Delta k_i \cdot \left(\lambda - \bar{\lambda}\right)^{n-1} + \frac{\partial \ln I}{\partial w}\sum_{i=1}^{n}\Delta w_i \cdot \left(\lambda - \bar{\lambda}\right)^{n-1}. (9)$$

**3. Results and Discussion**
We characterize the effect of including the PA ($\frac{\partial \ln I}{\partial p} \cdot \Delta p$) on ozone profile retrievals using both Super
Gaussian and standard Gaussian slit functions. Hereafter, the correction spectrum ($\frac{\partial \ln I}{\partial p}$) is derived using the
HR simulation. The PA coefficient ($\Delta p_i$) (one for each channel and for each order) is included as part of
the state vector to be iteratively and simultaneously retrieved with ozone. The a priori value is set to be zero



for all fitting coefficients, while the a priori error is set to be 0.1, empirically. We should note that the
empirical "soft calibration" is applied to OMI radiances before the spectral fitting, in order to eliminate the
wavelength and cross-track dependent systematic biases, due to the interference of the PA coefficients with
systematic measurement errors during the fitting process.

## 3.1 Characterization of the pseudo absorbers in ozone fitting procedure

Figure 4 displays how the zero-order PA coefficients ($\Delta p$) vary within one orbit when slit functions are
assumed as standard and Super Gaussians, respectively, along with variation of cloud fraction, surface
albedo, and cloud pressure from the retrievals. These fitting coefficients physically represent the difference
of slit parameters between radiance and irradiance in this implementation. Therefore, we normalize them
with the slit parameters derived from OMI solar irradiances for a better interpretation. Cross-track
dependent features are shown in slit width. The relative change of the slit width is more distinct in the UV1
band than in the UV2 band, whereas the change of the shape factor is more distinct in the UV2 band. The
UV2 slit widths increase typically within 5 % over the given spatial domain. However, the UV1 slit widths
increase from 10 % at most pixels up to 50 % at off-nadir positions in the high latitudes, which might be
caused by stray light differences between radiance and irradiance and intra-orbit instrumental changes. An
abnormal change of the UV1 slit parameters due to the scene heterogeneity is detected at the along-track
scan positions of ~300 and 900, respectively, where upper-level clouds are present. The UV2 shape factor
changes show a coherent sensitivity to bright surfaces under clear-sky condition over the northern high
latitudes. Fitting coefficients for the standard Gaussian show a quite similar spatial variation for the UV1
slit width (correlation = ~ 0.98), but an anti-correlation of ~ -0.62 for the UV2 slit width compared to those
for Super Gaussian due to the interference between shape factor and slit width.
Examples of the total PAs (eq. 9) are illustrated in Figure 5 when (a) zero and (b) first-order polynomial
are fitted, respectively. The UV1 total PA spectrum, regardless of which Gaussian is assumed as slit
function, is very similar because the spectral structure caused by the slit width change is dominant. It implies
that OMI ISRFs in the UV1 band are similar to the standard Gaussian, for both radiance and irradiance
measurements, consistent with the pre-launch characterization (Dirksen et al., 2006). However, in the UV2
band, the PA is mostly contributed from the shape factor change in the case of super Gaussian, and the total
PA spectrum is more noticeable for super Gaussian. Our results indicate that the PA for the shape factor
change is required to adjust the spectral structures due to the differences in the slit functions between
radiance and irradiance over the UV2 band. In the case of the wavelength dependent ISRF fit, the impact
of first-order PAs on OMI radiances is relatively visible in the wavelength range of 300-310 nm. This result
is physically consistent with the wavelength dependent property shown in the slit parameters derived from
OMI irradiances as shown in Figure 6 where slit parameters are characterized in 10-pixel increments





assuming the super Gaussian slit function. In UV1, the slit widths plotted as FWHM slightly decrease by ~
0.1 nm at shorter wavelengths than 288 nm, but more sharply vary by up to ~ 0.2 nm at longer wavelengths.
Compared to slit widths, the wavelength dependences of the shape factors are less noticeable, except at
boundaries of the window.  In the UV2 window, both slit width and shape factor are highly invariant.
**3.2 Impact of including pseudo absorbers on ozone profile retrievals**
Figures 7 to 10 evaluate the impact of including zero-order PAs on ozone profile retrievals. Figure 7
illustrates how different assumptions in the slit functions affect the ozone profile retrievals with respect to
the retrieval sensitivity and the fitting accuracy from the case shown in Figure 4. In this figure, the Degrees
of Freedom for Signal (DFS) represents the independent pieces of ozone information available from
measurements, which typically decreases as ozone retrievals are further constrained by other fitting
variables. The reduced DFS values (< 5 %) imply that the ozone retrievals are correlated slightly with PAs.
The fitting accuracy is assessed as the root mean square (RMS) of relative difference (%) between measured
and calculated radiances over the UV1 and UV2 ranges, respectively. Including the PAs makes little
difference in the UV1 fitting residuals for most of individual pixels (1-5 %), but significantly reduces
residuals in the UV2 range. The adjusted amount of the residuals with PAs are generally larger when
assuming super Gaussian slit functions. This comes from different assumptions for slit functions in deriving
soft calibration spectra, where slit functions were parameterized as standard Gaussians. Therefore, applying
soft calibration to OMI spectra entails somewhat artificial spectral structures if ISRFs are assumed as Super
Gaussian in ozone retrievals, and hence the impact of PAs on the spectral fitting becomes more considerable.
Figure 8 compares how the spectral residuals are adjusted with PAs when soft calibration is turned on and
off, respectively. Using super Gaussians causes larger amplitudes of the spectral fitting residuals than using
standard Gaussians, if soft calibration is turned on and PAs are excluded. On the other hand, some residuals
are reduced and more broadly structured if soft calibration is turned off. Including PAs eliminates/reduces
some spikes of fitting residuals as well as improves the consistency of the fitting accuracy between using
standard and super Gaussians at wavelengths above 300 nm.
The benefit of this implementation on ozone retrievals is further assessed through comparison with
Electrochemical Concentration Cell (ECC) ozonesondes collected from the WOUDC (https://woudc.org/)
and SHADOZ (https://tropo.gsfc.nasa.gov/shadoz/) networks during the period 2005 to 2008. We select 13
SHADOZ sites in the tropics and 38 WOUDC sites in the northern mid/high latitudes. The collocation
criteria is within +/- 1 ° in latitude and longitude and within 12 hours in time. For comparison, high-vertical
resolution (~100 nm) profiles of ozonesondes are interpolated onto OMI retrieval grids (~2.5 km thick).
We limit OMI/ozonesonde comparisons to OMI solar zenith angle < 85°, effective cloud fraction < 0.4,
surface albedo < 20 % (100 %) in tropics and mid-latitudes (high latitude), top altitude of ozonesondes >





30 km, ozonesonde correction factors ranging from 0.85 to 1.15 if they exist, and data gaps for each
ozonesonde no greater than 3km. Comparisons between OMI and ozonesondes are performed for the
tropospheric ozone columns (TCOs) over 3 different latitude bands and for ozone profiles including all the
sites, with and without PAs (zero-order) for standard and super Gaussian slit function changes, respectively.
Figure 9 shows the comparisons of tropospheric ozone columns as scatter plots. Without using PAs, the
retrievals show significant differences of (1.2-2.2 DU or 3.8-6.4%) especially in mean biases between super
and standard Gaussians, with negative biases of 0.2-0.7 DU for super Gaussians and positive biases of 0.8-
1.5 DU for standard Gaussians. Overall, OMI retrievals are in a better agreement with ozonesonde
measurements using super Gaussians. The correlations and standard deviations are very similar in the
tropics and mid-latitudes, but the retrievals with standard Gaussians show better correlation and smaller
standard deviations in high-latitudes. Consistent with Sun et al (2017), the retrievals show significant
differences between using standard and super Gaussians, although there are some inconsistencies in
comparing OMI and ozonesondes; the main inconsistent factors are listed as following: In this study, soft
calibration is turned on and a priori information is taken from TB climatology to perform OMI ozone profile
retrievals, whereas soft calibration is turned off and a priori information is taken from LLM climatology in
Sun et al. (2017). OMI/ozonesonde data filtering criteria are quite similar to each other, except that the
criteria of the solar zenith angle and cloud fraction are relaxed from 75° and 0.3 to 85° and 0.4, respectively,
and the adjustment of ozonesondes with correction factor given for the WOUDC dataset is turned on in this
study. Comparison is performed by latitudes here whereas global comparison is analyzed in Sun et al.
(2017). After accounting for the slit differences between radiances and irradiances using PAs, the retrievals
are significantly improved for both standard and super Gaussians and these two retrievals become consistent
except for the use of super Gaussians in the tropics. The mean biases in the tropics and mid-latitudes are
almost eliminated, to within 0.3 DU, but the standard deviations and correlation do not change much,
slightly worse in the tropics and better in the mid-latitudes. In the high-latitudes, the standard deviations
and correlation are significantly improved especially for using super Gaussians, but the mean biases are
similar to the standard Gaussian without PAs. The lack of improvement with PAs in the tropics with super
Gaussians illustrates that ISRFs of radiances are quite similar to those of irradiances in the tropics, while
super Gaussians better parameterize OMI ISRFs than standard Gaussians. This is consistent with the
comparison of the fitting accuracy of the UV2 band as shown in Figure 7, where the fitting residuals are
slightly reduced in the tropics when super Gaussians are linearized, but the standard Gaussian linearization
significantly improves the fitting accuracy. The mean biases of the profile comparison as shown in Figure
10 clearly shows that including PAs to account for ISRF differences significantly reduces mean biases
below 10 km and the general altitude dependence and improves the consistency between using standard
and super Gaussians; the standard deviations also show noticeable improvement in the altitude range of 10-



20 km for both Gaussians. The significant improvement at all latitudes corroborates the change of ISRFs
between radiance and irradiance along the orbit as conjectured by Sun et al. (2017). The consistency
between using standard and super Gaussians after using PAs is mainly because there is strong anti-
correlation between the slit width and shape partial derivatives as shown in Figure 2, so the adjustment of
slit width only in the use of standard Gaussian can achieve almost the same effect as the adjustment of both
parameters in the use of super Gaussian. Accounting for the wavelength dependent change of the ISRFs
with first-order PAs makes insignificant differences to both fit residuals and ozone retrievals (not shown
here). This could be mainly explained with the fact of the negligible wavelength dependence of OMI ISRFs
especially in UV2 as shown in Figure 5 where the PA spectrum ($\frac{\partial \ln I}{\partial p} \cdot \Delta p$) shows almost no variance,
except at the upper boundary of the UV1 as well as in Figure 6 where the UV2 slit parameters derived from
irradiances in the sub-fit windows vary within 0.05 nm for FWHM and 0.2 for shape factor.

## 4. Summary

The knowledge of the Instrument Spectral Response Functions (ISRFs) or slit functions is important
for ozone profile retrievals from the Hartley and Huggins bands. ISRFs can be measured in the laboratory
prior to launch, but they have been typically derived from solar irradiance measurements assuming
Gaussian-like functions in order to account for the effect of the ISRF changes after launch. However, the
parameterization of the ISRFs from solar irradiances could be inadequate for achieving a high accuracy of
the fitting residuals as ISRFs in radiances could significantly deviate from those in solar radiances (Beirle
et al., 2017) and might affect ozone profile retrievals as suggested in Sun et al. (2017). Therefore, this study
implements a linearization scheme to account for the spectral errors caused by the ISRFs changes as Pseudo
Absorbers (PAs) in an optimal estimation based fitting procedure for retrieving ozone profiles from OMI
BUV measurements using the SAO ozone profile algorithm. The ISRFs are assumed to be the generic super
Gaussian that can be used as standard Gaussian when fixing the shape factor to 2. This linearization was
originally introduced in Beirle et al. (2017) for DOAS analysis, but this study extends this application and
more detail how to implement in practice using two different approaches to derive radiance errors from slit
function partial derivatives with respect to slit parameters. These two approaches correspond to the two
methods of simulating radiances at instrument spectral resolution, one using effective cross sections which
were previously used in the SAO ozone profile algorithm and are still used in most of the trace gas retrievals
from the UV and visible, and the other calculating radiances at high resolution before convolution, which
is the preferred method in the SAO ozone profile algorithm. Consistent PAs are derived with these two
approaches, as expected.




The fitting coefficients ($\triangle$ p) to the PAs, representing the difference of slit parameters between radiance
and irradiance, are iteratively fitted as part of the state vector along with ozone and other parameters. The
UV1 slit parameters show distinct cross-track-dependent differences, especially in high-latitudes. In
addition, an abnormal $\triangle$ p caused by scene heterogeneity is observed around bright surfaces and cloudy
scenes. The total PA spectrum ($\frac{\partial I}{\partial p} \cdot \triangle$ p) illustrates that the slit width change causes most of the spectral
structures in the UV1 band because the OMI ISRFs are close to Gaussian. Otherwise, the ISRF change
results into different spectral responses in the UV2 band with different Gaussian functions because the
adjustment of the shape factor becomes more important in accounting for the convolution error when using
super Gaussians.
Insignificant wavelength dependence on OMI slit functions is demonstrated from slit function
parameters derived from irradiances in the sub-fit window, which leads to little difference in ozone profile
retrievals when zero and first-order wavelength dependent PA coefficients are implemented to fit the
spectral structures caused by slit function errors, respectively. Therefore we evaluate the benefit of
including the zero-order PAs fit on both the accuracy of the fitting residuals and the quality of retrieved
ozone profiles through validation against ozonesonde observations. Some spikes in the fitting residuals are
reduced or eliminated. Commonly, including PAs makes little change on both fit residuals and ozone
retrievals in the tropics if a super Gaussians are assumed as ISRFs but this is not the case for the standard
Gaussian assumption. Retrievals using standard and super Gaussians agree better if slit function errors are
accounted for by including PAs. Using PAs ultimately demonstrates substantial improvement of ozone
profile retrievals in the comparison of tropospheric ozone columns and ozone profiles up to 30 km. Using
super Gaussians, the TCO comparison shows significant improvement in mean biases in mid-latitudes and
in standard deviations in high-latitudes. Using standard Gaussians, the TCO comparison also shows
significant improvement in mean biases in the tropics. The profile comparison generally shows
improvement in mean biases as well as in standard deviation in the altitude range 10-20 km. More
importantly, using these PAs make the retrieval consistent between standard and super Gaussians. Such
consistency is due to the anti-correlation between slit width and shape PAs. This study demonstrates the
slit function differences between radiance and irradiance and its usefulness to account for such differences
on the pixel-to-pixel basis. In this experiment, the soft spectrum, derived with the standard Gaussian
assumption, is applied to remove systematic measurement errors before spectral fitting, indicating that the
evaluation of ozone retrievals might be unfairly performed for the super Gaussian function implementation.
Nonetheless, OMI ozone profile retrievals show better agreement with ozonesonde observations when the
super Gaussian is linearized. Actually, the fitting residuals are slightly more broadly structured with super
Gaussians than with standard Gaussians if the soft-calibration and PAs are turned off, indicating the benefit





of deriving a soft calibration with the super Gaussians. Therefore, there is still room for achieving better
benefits when using the PAs on ozone profile retrievals by applying the soft calibration derived with super
Gaussians.

**Acknowledgement**

We acknowledge the OMI science team for providing their satellite data and the WOUDC and SHADOZ
networks for their ozonesonde datasets. Research at the Smithsonian Astrophysical Observatory by J. Bak,
X. Liu, K. Sun, and K. Chance was funded by NASA Aura science team program (NNX14AF16G &
NNX17AI82G).

Azam, F. and Richter, A.: GOME2 on MetOp: Follow on analysis of GOME2 in orbit degradation, Final
report,        EUM/CO/09/4600000696/RM,        2015,        available        at:        http://www.doas
bremen.de/reports/gome2_degradation_follow_up_final_report.pdf (last access: 7 September 2016),
2015.

Bak, J., Liu, X., Wei, J. C., Pan, L. L., Chance, K., and Kim, J. H.: Improvement of OMI ozone profile
retrievals in the upper troposphere and lower stratosphere by the use of a tropopause-based ozone profile
climatology, Atmos. Meas. Tech., 6, 2239–2254, doi:10.5194/amt-6-2239-2013, 2013.

Bak, J., Liu, X., Kim, J.-H., Haffner, D. P., Chance, K., Yang, K., and Sun, K.: Characterization and
correction of OMPS nadir mapper measurements for ozone profile retrievals, Atmos. Meas. Tech., 10,
4373-4388, https://doi.org/10.5194/amt-10-4373-2017, 2017.

Beirle, S., Sihler, H., and Wagner, T.: Linearisation of the effects of spectral shift and stretch in DOAS
analysis, Atmos. Meas. Tech., 6, 661–675, doi:10.5194/amt-6-661-2013, 2013.

Beirle, S., Lampel, J., Lerot, C., Sihler, H., and Wagner, T.: Parameterizing the instrumental spectral
response function and its changes by a super-Gaussian and its derivatives, Atmos. Meas. Tech., 10, 581-
598, https://doi.org/10.5194/amt-10-581-2017, 2017.

Brion, J., Chakir, A., D. Daumont, D., and Malicet, J.: High-resolution laboratory absorption cross section
of O3. Temperature effect, Chem. Phys. Lett., 213(5–6), 610– 612, 1993.

Cai, Z., Liu, Y., Liu, X., Chance, K., Nowlan, C. R., Lang, R., Munro, R., and Suleiman, R.: ,
Characterization and correction of Global Ozone Monitoring Experiment 2 ultraviolet measurements
and application to ozone profile retrievals, J. Geophys. Res., 117, D07305, doi:10.1029/2011JD017096,
2012.

Caspar, C. and Chance, K.: GOME wavelength calibration using solar and atmospheric spectra, Third ERS
Symposium on Space at the Service of our Environment, Florence, Italy, 14–21 March, 1997.

Dobber, M., Voors, R., Dirksen, R., Kleipool, Q., and Levelt, P.: The high-resolution solar reference
spectrum between 250 and 550 nm and its application to measurements with the Ozone Monitoring
Instrument, Solar Physics, 249, 281–291, 2008.Kim, P. S., Jacob, D. J., Liu, X., Warner, J. X., Yang,



K., Chance, K., Thouret, V., and Nedelec, P.: Global ozone–CO correlations from OMI and AIRS:
constraints on tropospheric ozone sources, Atmos. Chem. Phys., 13, 9321-9335,
https://doi.org/10.5194/acp-13-9321-2013, 2013.
Kim, P. S., Jacob, D. J., Liu, X., Warner, J. X., Yang, K., Chance, K., Thouret, V., and Nedelec, P.: Global
ozone–CO correlations from OMI and AIRS: constraints on tropospheric ozone sources, Atmos. Chem.
Phys., 13, 9321-9335, https://doi.org/10.5194/acp-13-9321-2013, 2013.
Levelt, P. F., van den Oord, G. H. J., Dobber, M. R., Malkki, A., Visser, H., de Vries, J., Stammes, P.,
Lundell, J. O. V., and Saari, H.: The ozone monitoring instrument, IEEE Transactions on Geoscience
and Remote Sensing, 44, 1093–1101, doi:10.1109/TGRS.2006.872333, 2006.
Liu, X., Chance, K., Sioris, C. E., Spurr, R. J. D., Kurosu, T. P., Martin, R. V., and Newchurch, M. J.:
Ozone profile and tropospheric ozone retrievals from Global Ozone Monitoring Experiment: algorithm
description and validation, J. Geophys. Res., 110, D20307, doi: 10.1029/2005JD006240, 2005.
Liu, X., Bhartia, P.K, Chance, K, Spurr, R.J.D., and Kurosu, T.P.: Ozone profile retrievals from the ozone
monitoring instrument. Atmos. Chem. Phys., 10, 2521–2537, 2010.
Liu, C., Liu, X., Kowalewski, M.G., Janz, S.J., González Abad, G., Pickering, K.E., Chance, K., and
Lamsal., L.N.: Characterization and verification of ACAM slit functions for trace gas retrievals during
the 2011 DISCOVER-AQ flight campaign, Atmos. Meas. Tech., 8, 751-759, doi:10.5194/amt-8-751-
2015, 2015.

Sioris, C. E. and Evans, W. F. J.: Impact of rotational Raman scattering in the $O_2$ A band, Geophys. Res.
Lett., 27, 4085–4088, 2000.
Spurr, R. J. D.: Linearized pseudo-spherical scalar and vector discrete ordinate radiative transfer models
for use in remote sensing retrieval problems, in: Light Scattering Reviews, edited by: Kokhanovsky, A.,
Springer, New York, 2008.
Sun, K., Liu, X., Huang, G., González Abad, G., Cai, Z., Chance, K., and Yang, K.: Deriving the slit
functions from OMI solar observations and its implications for ozone-profile retrieval, Atmos. Meas.
Tech., 10, 3677-3695, https://doi.org/10.5194/amt-10-3677-2017, 2017.
van Hees, R. M., Tol, P. J. J., Cadot, S., Krijger, M., Persijn, S. T., van Kempen, T. A., Snel, R., Aben, I.,
and Hoogeveen, Ruud W. M.: Determination of the TROPOMI-SWIR instrument spectral response
function, Atmos. Meas. Tech., 11, 3917-3933, https://doi.org/10.5194/amt-11-3917-2018, 2018.
Wilmouth, D. M., Hanisco, T. F., Donahue, N. M., and Anderson, J. G.: Fourier transform ultraviolet
spectroscopy of the $A2II_{3/2} - X^2II_{3/2}$ Transition of BrO, J. Phys. Chem. A., 103(45), 8935– 8945, 1999.








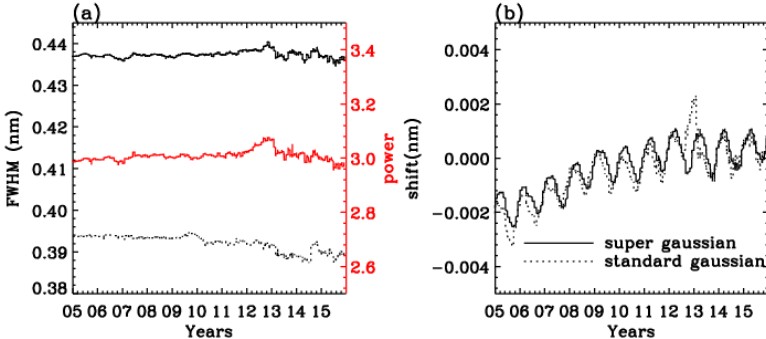


**Figure 1. Time series of (a) slit parameters and (b) wavelength shifts for OMI daily irradiance measurements (310-330 nm) at nadir cross track position when Super Gaussians (solid line) and standard Gaussians (dotted line) are parameterized as slit function shapes, respectively.**









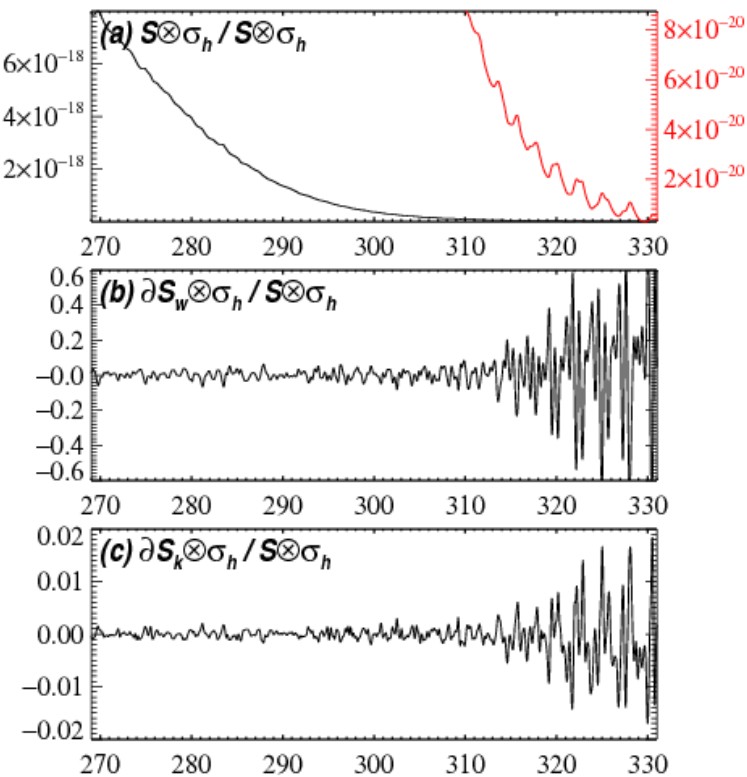


**Figure 2. (a) Ozone absorption cross sections (cm²/molecule) ($\delta_h$) at different scales (red and black) at a representative temperature (238.12 K) calculated via convolution of high-resolution (0.01 nm) reference spectrum with the Super Gaussian slit function, S ($k = 2.6, w = 0.26$ nm). (b) and (c) its derivatives with respect to slit parameters ($\partial S_p = \frac{\partial S}{\partial p}$), $w$ and $k$, respectively, normalized to the convolved cross sections.**

463

464



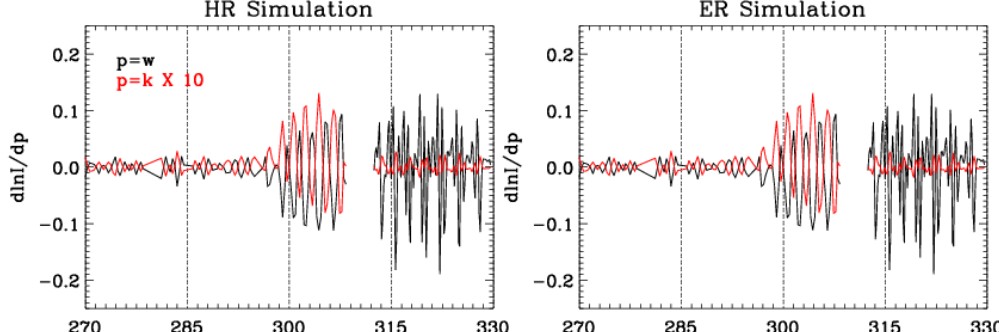

**Figure 3. Derivatives of OMI radiance spectrum simulated using high-resolution (HR) and effective resolution (ER) cross section spectra with respect to slit parameters assuming a Super Gaussian function. $\mathrm{dlnI}/\mathrm{d}k$ is multiplied by a factor of 10 to visually match $\mathrm{dlnI}/\mathrm{d}w$ in the same y-axis.**



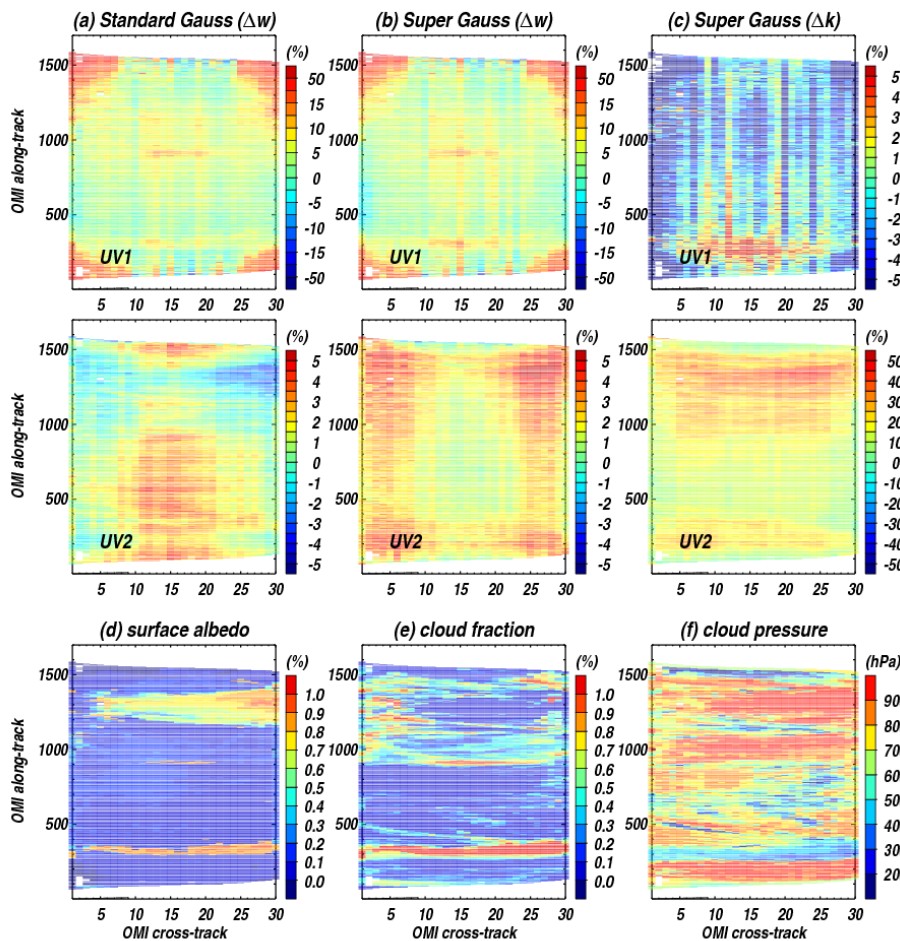

469

**Figure 4.** Pseudo absorption coefficients ($\Delta w, \Delta k$) for fitting the OMI radiances due to slit function changes assuming (a) standard Gaussian and (b-c) Super Gaussian, within the first orbit of measurements on 1 July 2006, with (d-f) the corresponding geophysical parameters. $\Delta w$ and $\Delta k$ is displayed after being normalized with $w_o$, and $k_o$, the slit parameters derived from OMI solar irradiance measurements.



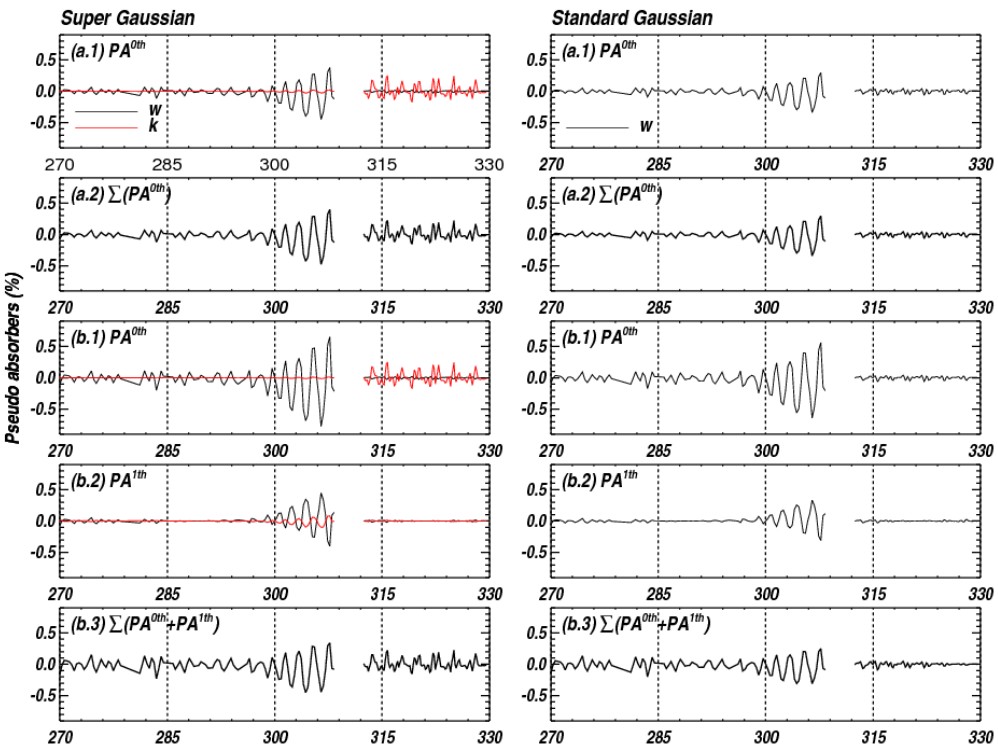

475

**Figure 5. (a.1) Pseudo absorber spectra ($\frac{\partial lnI}{\partial p} \times \Delta p$ ) for zero order slit parameters and (a.2) its total**

**spectra for (left) Super Gaussian and (right) Standard Gaussian function parameterizations, respectively.**

**(b) Same as (a), but for first order polynomial fit. The case represents an average at nadir in the latitude**

**zone 30°-60°N.**









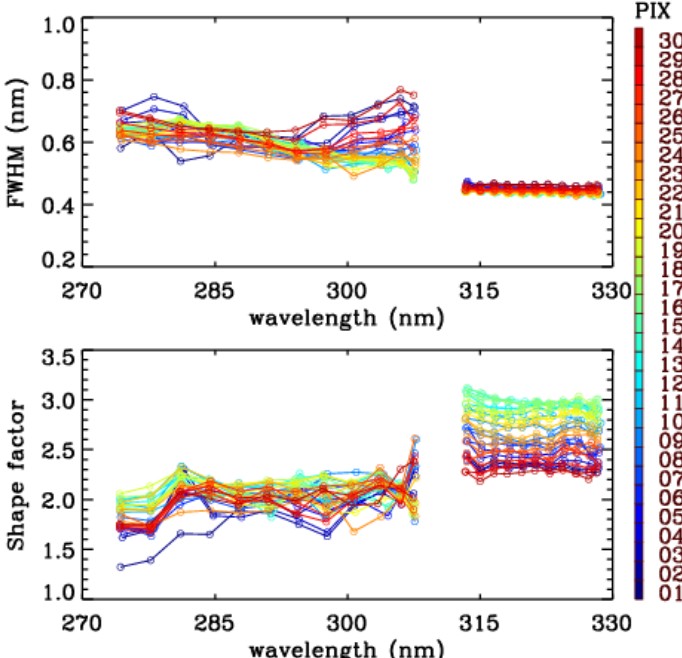


**Figure 6. OMI ISRF FWHM (nm) and shape factor ($k$) as functions of the center wavelength, as derived**
**from OMI solar irradiances assuming Super Gaussian functions over a range of 31 spectral pixels in 10-**
**pixel increments. Different colors represent different cross-track positions from 1 (blue) to 30 (red).**





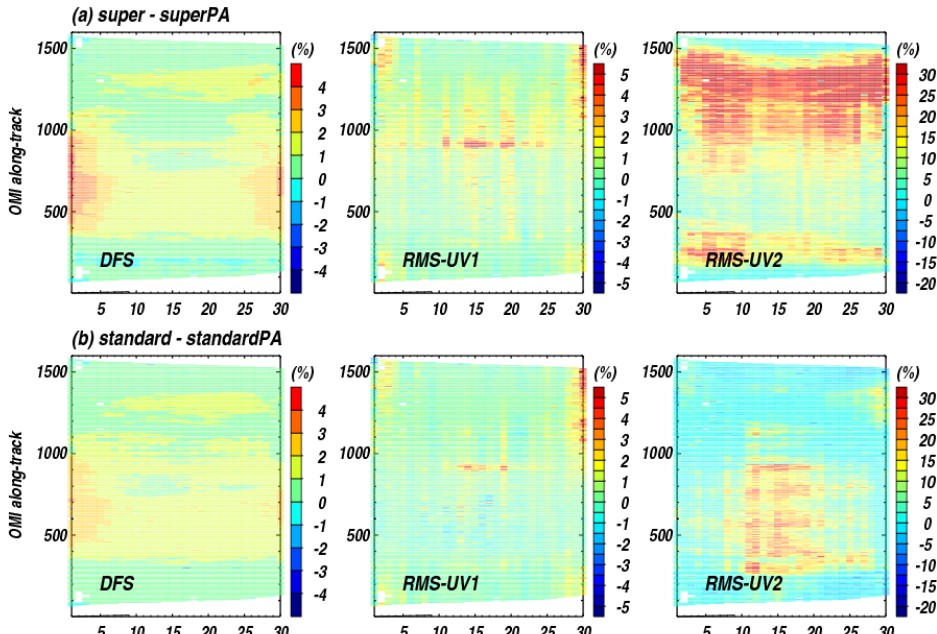

**Figure 7. Same as Figure 4, but for comparisons of the Degrees of Freedom for Signal (DFS) and the Root Mean Square (RMS) of spectra fitting residuals in UV 1 and UV2 with and without zero-order pseudo absorber. Positive values indicate that both fitting residuals and DFSs are reduced due to the pseudo absorber.**





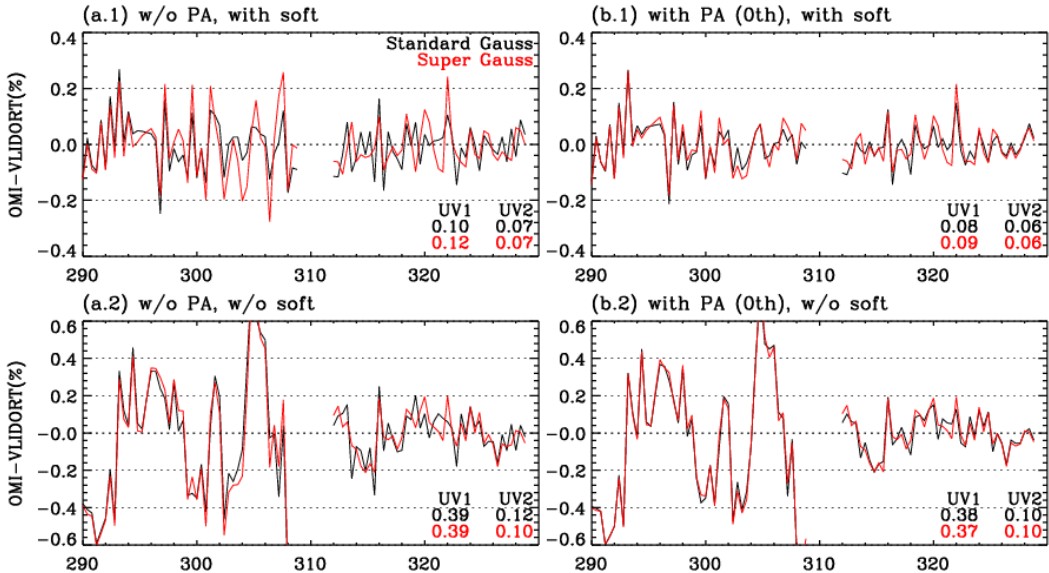

**Figure 8. Average differences (%) between measured (OMI) and simulated (VLIDORT) radiances (residuals) at the nadir cross-track pixel in the tropics (30°S-30°S) without (a) and with (b) zero-order pseudo absorbers (PA) when the standard Gaussian (black line) and the Super Gaussian (red line) are assumed as ISRFs, respectively. Upper/lower panels represent the fit results with soft calibration being turned on/off. The residuals in the UV1 (< 310 nm) are scaled by a factor of 2 to fit in the given y-axis. In the legend, the RMS of residuals (%) are given for UV1 and UV2 wavelength ranges, respectively.**



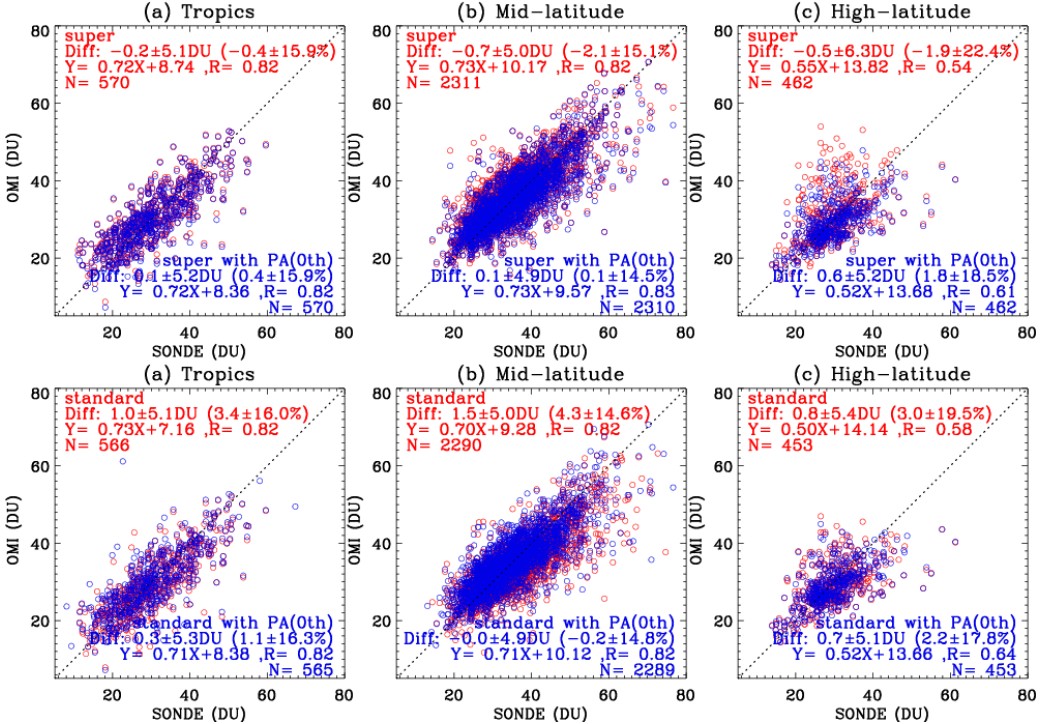

**Figure 9.** Comparison of OMI and ozonesonde tropospheric column ozone over (a) the tropics (30°S-30°N), (b) mid-latitudes (30°N-60°N), (c) high-latitudes (60°N-90°N), with different slit function assumptions/implementations. Super and standard Gaussians are assumed as slit function for the upper and lower results, respectively. Different colors represent the implementations with (blue) and without (red) pseudo absorbers.





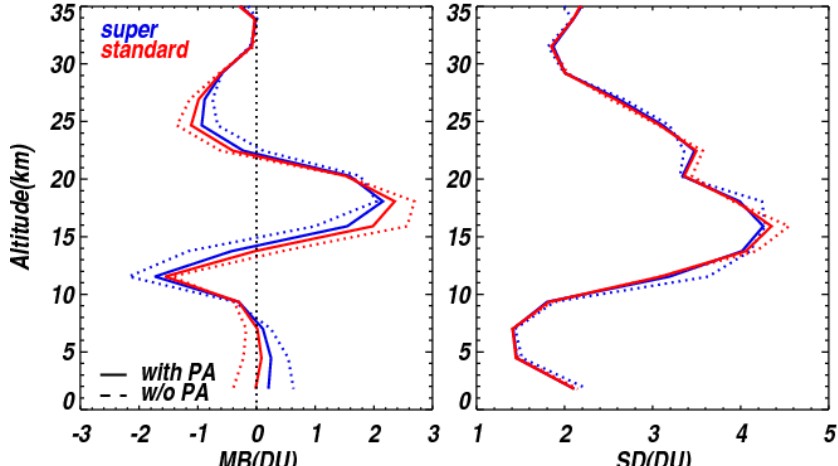


**Figure 10. Global mean biases at each OMI layer and 1 σ standard deviations of the differences between**
**OMI and ozonesondes, with different slit function assumptions/implementations.**



