# Peer review of "Linearization of the effect of slit function changes for improving OMI ozone profile retrievals"

_Atmospheric Measurement Techniques, 2019_

## Referee Comment (RC1) · Anonymous Referee #1 · 3 May 2019

The paper applies a linearisation of the ISRF for the retrieval of ozone profiles from OMI measurements. The linearisation approach was introduced by Beirle et al., 2017 (BE17 hereafter), which is referenced appropriately. However, the authors should generally specify more clearly which steps are adopted from BE17, and what are original/new ideas/methods/results of their study.

The adaptation of the ISRF parameterization for radiances seems to be new and interesting. However, there are some complications which have to be investigated in detail and discussed thoroughly. I recommend publication in AMT after these major revisions have been made.

1. Irradiance vs. radiance

BE17 presented the ISRF parameterization for a fit of a measured irradiance to a high-resolution solar atlas. In the current study, the authors apply the parameterization to radiances. This implies that the PAs depend on the Ozone column, and the spectral structures are different for each satellite pixel! This is not clearly stated in the manuscript and should be quantified (i.e. compare the PAs for high/medium/low ozone). Other absorbers have the same effect, i.e. the spectral patterns of the PAs depend e.g. on the strength of the Ring effect (thus on clouds!). This has to be discussed.

2. The abstract is contains some statements which are not supported by the presented data:

a) Abstract, first sentence: "reduces the spectral fit residuals caused by the slit function errors". Please add a figure of the spectral analysis with and without PAs in order to substantiate this statement.

b) End of abstract: "Comparisons with ozonesondes demonstrate substantial improvements with the use of PAs". In fig. 10, I see almost no difference, and particularly no "substantial improvements", no matter which function is used nor whether PAs are included or not. Obviously, there are systematic differences remaining compared to Ozone sondes which are not related to the ISRF parameterization.

3. How do the derived ISRFs look like, and how do they compare to the prelaunch measurements performed for OMI?

Fig. 5: What is the meaning of the sum of PAs? Each PA has to be scaled by the respective Delta p. Thus the spectral patterns must not just be added!?

Fig. 5: The 1st order spectra look wrong. According to Eq. 9, they are 0 in the center of the wavelength window and increase towards the edges (compare Fig. 10 in BE17). The presented spectra look the other way round.

Fig. 9: Specify the time range of the presented data.

Fig. 10: The unit on the x axis must be DU per km or per vertical layer. Please specify.

[Figure]

Fig. 10: Abbreviation "MB" is not defined.

[Figure]

---

## Referee Comment (RC2) · Anonymous Referee #2 · 8 May 2019

This paper is well organized to describe a methodology for reducing the spectral fit residuals. The subject of the paper is appropriate to AMT. Below are a few comments concerning clarifications / extensions for consideration in the final publication in AMT.

Major comments [1] The PROFOZ algorithm applies the pre-estimated, pixel dependent "soft calibration" factors to the normalized radiances, while conducting the ozone profile retrievals. The "soft calibration" factors seem, by design, accounting for the imperfectness of OMI L1B earth shine radiances and solar irradiances calibration, parameterization of the pixel & wavelength dependent ISRFs, and forward model parameters (absorption cross-sections, surface albedo) etc. The PROFOZ also fits scaling factors for the pre-estimated mean spectral fit residuals (Liu 2010 a, b) for UV1 and UV2 bands accordingly, to account for the remaining systematic errors that were not fully removed

from "soft calibration" process. This work suggests fit additional ISRF PA coefficients is necessary for OMI ozone profile retrievals. It seems there might some degeneracy among these approaches. The authors should elaborate whether employing pixel & temporal dependent 'soft calibration' factors, or fitting the mean spectral residuals could also achieve the goals same to employing the presented PA approach, in terms of reducing the spectral fit residuals. Are the Jacobians of these PA coefficients, orthogonal to the pre-estimated mean fitting residual spectra, or any other Jacobians of parameters in the retrieval vector?

[2] The authors should obtain time serials of retrieved ISRF PA coefficients. Do they show trends similar to Figure 1? At least for Nadir pixel, if not all pixels.

[3] The authors evaluated the impacts of with/without retrieving PA coefficients on the bias/RMS between retrieved ozone and in-situ ozonesonde measurements (Figure 9). However, the evaluation only made for the period of 2005 to 2008, when OMI instrument was within design lifetime. The authors should also evaluate the performances using the satellite-ozonesonde measurements in other time periods including 2010 and 2012-2013, when the ISRF characteristics were significantly different than the earlier years, as shown in Figure 1. The authors should also add some discussions on the possible reasons causing these sharp changes of ISRF characteristics.

Technical comments

[1] Have the authors evaluated the impacts of this methodology on the L2 retrieval throughput/yields?

[2] Line 29, use the statistical numbers on the bias/RMS differences to replace the word "substantial".

[3] Line 47, the authors should consider to revise "by narrow and weak absorption features of the temperature-dependent Huggins bands (320-360 nm)" to "by the 320-330 nm absorption features residing in the temperature-dependent Huggins bands.",

since neither this work nor the referenced studies utilized spectral region > 330 nm in the OMI ozone profile retrievals. "narrow and weak" are general terms and might subjective, e.g., this statement will break down. When comparing within the Chappuis bands, the refereed portion of Huggins bands (>320 nm) is no longer weak.

[4] Line 50, I will suggest to cite the following studies on OMI ozone profile retrievals, since [1] they made use of the ISRFs from Dirksen et al., [2006] cited a few times in this work, [2] the quality evaluation have been conducted by the comparison with in-situ ozonesonde measurements, [3] same to Liu et al., 2010 cited in this work, these studies were conducted prior to the era of including PA coefficients in the retrieval vector.

Kroon, M., de Haan, J. F., Veefkind, J. P., Froidevaux, L., Wang,R., Kivi, R., and Hakkarainen, J. J.: Validation of operationalozone profiles from the Ozone Monitoring Instrument, J. Geo-phys. Res., 116, D18305, doi:10.1029/2010JD015100, 2011.

Mielonen, T., de Haan, J. F., van Peet, J. C. A., Eremenko, M., andVeefkind, J. P.: Towards the retrieval of tropospheric ozone withthe Ozone Monitoring Instrument (OMI), Atmos. Meas. Tech.,8, 671–687, https://doi.org/10.5194/amt-8-671-2015, 2015.

Fu, D., Kulawik, S. S., Miyazaki, K., Bowman, K. W., Worden, J. R., Eldering, A., Livesey, N. J., Teixeira, J., Irion, F. W., Herman, R. L., Osterman, G. B., Liu, X., Levelt, P. F., Thompson, A. M., and Luo, M.: Retrievals of tropospheric ozone profiles from the synergism of AIRS and OMI: methodology and validation, Atmos. Meas. Tech., 11, 5587-5605, https://doi.org/10.5194/amt-11-5587-2018, 2018.

Fu, D., Worden, J. R., Liu, X., Kulawik, S. S., Bowman, K. W., and Natraj, V.: Characterization of ozone profiles derived from Aura TES and OMI radiances, Atmos. Chem. Phys., 13, 3445-3462, https://doi.org/10.5194/acp-13-3445-2013, 2013.

[5] Line 60, might be a typo (radiance repeated twice)?. Do the authors mean "differences in stray light between radiance and irradiance" or "differences in stray light among OMI measurements"?

[6] Line 61, It seems that "intra-orbit instrumental changes" is duplicating the statement of "the instrument temperature change". Please clarify (or remove one).

[7] Figures 1, 2, 4, 5, 7, 8, 9 and 10, increase the tick length for improving their visibility.

[8] Figures 5, 6, 8, 9 and 10 captions, state the date/time range of the data presented in the figures. It is not where they are all for 1 July 2006, shown in Figure 4 caption.

[9] Figure 9, create a table and move the statistical values to the table. Having all these numbers on the plots resulted in the plots being too busy to read.

[10] Figure 10 # please spell out the "MB" and "SD" in the x axis title, - space suffice to hold the full name and they were not defined in the caption. # Add two panels to show the differences among data sets, as a function of altitude?

[11] Finally, please keep the 'style' of all figures in a similar fashion. e.g., the panel index of Figure 2 (a), (b) and (c) are inside the plots, while the other figures are outside of the plots. I understand that there is no space for the subtitles outside Figure 2b and 2c, due to the x axis labels. The authors should consider to remove those x axis labels, since all panels could share the one of panel c. Similarly, there are unnecessary axis labels in other figures, e.g., Figures 4, 5, 6, 7, 8, 9, and 10, when some subpanels having an identical scale/range across a row and/or a column, authors should consider remove the unnecessary labels in x or y axis, to help readers easily catch key information presented in the figures.

---

## Author Comment (AC1) · 26 May 2019

The author would like to thank Anonymous referee #1 for the constructive and helpful suggestions on this manuscript.
We replied to 1 general comment and 8 specific comments.

**General Comment**

**C1.** The paper applies a linearization of the ISRF for the retrieval of ozone profiles from OMI measurements. The linearization approach was introduced by Beirle et al., 2017 (BE17 hereafter), which is referenced appropriately. However, the authors should generally specify more clearly which steps are adopted from BE17, and what are original/new ideas/methods/results of their study. The adaptation of the ISRF parameterization for radiances seems to be new and interesting. However, there are some complications which have to be investigated in detail and discussed thoroughly. I recommend publication in AMT after these major revisions have been made.

**R1.** According to this comment, we have specified what this paper adopted from BE17 and what we advanced in implementing the slit function linearization in Section 2.2, as following " In Beirle et al. (2017) a slit function linearization was implemented only to fit solar irradiances from GOME-2. We implement the slit function linearization to fit radiances in the SAO ozone profile algo rithm (Liu et al. 2010), (Liu et al. 2010). ~ In DOAS analysis, the pseudo absorber is define d as $\frac{\partial S}{\partial p} \otimes \sigma_h$ ( $\sigma_h$ is a high-resolution absorption cross section), which could be calculated at a computationally low-cost. In our optimal estimation based ozone profile retrievals, it is conc eptually defined as $\frac{\partial S}{\partial p} \otimes I_h$ ( $I_h$ is a high-resolution simulated radiance), which is computationa lly very expensive because of on-line radiative calculation for a ~ 60 nm wide fit window on the spatial pixel-to-pixel basis. We now introduce how to implement the slit function lineariza tion to derive the derivatives of the OMI radiances with respect to slit function changes in t wo different radiative transfer approaches used in the SAO ozone profile algorithm, i.e., the e ffective cross section approach in Liu et al (2010) and the updated high-resolution convolutio n approach described in Kim et al. (2013), respectively."

**Specific Comments**

**C1.** Irradiance vs. radiance. BE17 presented the ISRF parameterization for a fit of a measured irradiance to a high-resolution solar atlas. In the current study, the authors apply the parameterization to radiances. This implies that the PAs depend on the Ozone column, and the spectral structures are different for each satellite pixel! This is not clearly stated in the manuscript and should be quantified (i.e. compare the PAs for high/medium/low ozone). Other absorbers have the same effect, i.e. the spectral patterns of the PAs depend e.g. on the strength of the Ring effect (thus on clouds!). This has to be discussed.

**R1. -** Yes, PAs vary with each satellite pixel. We plotted PAs with respect to slit width for 138 different satellite pixels (S1). The amplitude of PAs increases with latitude/solar zenith angle, but the spectral structures do not change because it arises from errors due to the convolution process of high-resolution absorption cross sections dominated by ozone. This discussion has been included in the revised manuscript, "The amplitude of $\frac{dlnI}{dp}$ varies with different satellite pixels (e.g., ozone profile shape, geometry, and cloud/surface property), but the spectral peak positions do not change because they arise from the errors due to the convolution process of

[Figure]

**S1. dlnI/dw for 138 pixels at cross-track =15, 0<lat<80, sza<80, and cloud fraction <0.1. The difference colors represent from lower latitudes at red color to higher latitudes at blue color.**

high-resolution absorption cross-sections dominated by ozone." at line 211.

- As this review pointed out, other elements of the state vector also have some correlation with cloud fraction, surface albedo, cross track position (e.g. UV1 radiance/ozone cross section shift, UV2 ring scaling parameter, UV1 radiance/irradiance shift). However, it is complex to figure out how these state vectors are interacting with PA coefficients because of weak correlation (<+/- 0.3 for UV1 variables and <+/- 0.1 for UV2 variables) between their jacobians. The PAs are not directly dependent on the strength of the Ring effect in the current implementation, because Ring effect is not fully coupled with the VLIDORT, but calculated using a first-order single scattering model and then scaled with a polynomial to be fitted.

[Figure]

**S2. Same as Fig. 4, but for other state vectors.**

**C2.** The abstract contains some statements which are not supported by the presented data:

a) Abstract, first sentence: "reduces the spectral fit residuals caused by the slit function errors". Please add a figure of the spectral analysis with and without PAs in order to substantiate this statement.

b) End of abstract: "Comparisons with ozonesondes demonstrate substantial improvements with the use of PAs". In fig. 10, I see almost no difference, and particularly no "substantial improvements", no matter which function is used nor whether PAs are included or not. Obviously, there are systematic differences remaining compared to Ozone sondes which are not related to the ISRF parameterization.

**R2-a.** Figs 7 and 8 support the benefit of including a pseudo absorber to improve the fit accuracy. Figure 7 compares the root mean square (RMS) of relative difference (%) between measured and calculated radiances over the UV1 and UV2 ranges, respectively. Including the PAs makes little difference in the UV1 fitting residuals for most of individual pixels (1-5 %), but significantly reduces residuals in the UV2 range (10-25%). In Figure 8, the spectral fit residuals are compared with and without PAs, indicating that including PAs eliminates/reduces some spikes of fitting residuals as well as improves the consistency of the fitting accuracy between using standard and super Gaussians at wavelengths above 300 nm. But as the reduction in the fitting residuals compared to the overall magnitude of the residuals is small, I modify "reduces the spectral fit residuals caused by" to "accounts for"

**R2-b.** "Substantial" is replaced with "noticeable". It typically reduces the mean biases with relative to ozonesonde and significantly reduces the standard deviations at high latitudes in the case of super Gaussian. It also makes the mean biases consistent at different latitudes and between the use of standard Gaussian or super Gaussian. In Fig. 10, we think that the benefit of applying ISRF on comparison is not negligible. This figure is re-plotted below in the unit of % and added in the revised manuscript, showing that ~ 5 % of mean biases is eliminated by PA in the lower troposphere. Furthermore, including PAs clearly makes the retrievals consistent between standard and super Gaussians from up to 10% to within 2%.

[Figure]

**S3. Comparison of relative differences (%) between OMI and ozonesonde as a function of altitude, with different slit function assumption and implementation.**

**C3.** How do the derived ISRFs look like, and how do they compare to the prelaunch measurements performed for OMI?

**R3.** The comparisons between pre-flight ISRF measurements and the derived ISRFs from solar irradiances are detailed in Sun et al. (2017). In Sun et al. (2017) and this study, the ISRFs are parameterized as a super Gaussian or standard Gaussian from solar irradiance measurements, which are used to convolve high-resolution cross-section spectra into OMI spectral resolution for radiative transfer calculation. In this study, we furthermore focused on implementing the slit function linearization to account for the spectral structures caused by the ISRF difference between radiance and irradiance. A fitting parameter is included as a state vector to adjust the amplitude of this spectral structure with each different pixel. This parameter is named by "pseudo absorber coefficient", which physically represents not directly ISRFs, but the deviation of ISRFs in radiances from those in solar measurements. ISRFs deviates temporally and spatially and thereby it is complex to represent the ISRFs in radiances.

**C4.** Fig. 5: What is the meaning of the sum of PAs? Each PA has to be scaled by the respective Delta p. Thus the spectral patterns must not just be added!?

**R4.** The sum of PAs indicates the total spectral structures caused by the slit function errors. Yes, the PAs cannot be just added together; they will be scaled by the PA coefficients before added together. To avoid the confusion, we have declared it as "the sum of PAs multiplied by corresponding PA coefficients".

- (Caption in Fig.5) Figure 5. (a.1) Pseudo absorber spectra multiplied by corresponding zero order coefficients, $\frac{\partial lnI}{\partial p} \times \Delta \boldsymbol{p_o}$ and (a.2) the sum of them for (left) super Gaussian and (right) standard Gaussian function parameterizations, respectively.

- (Line 250) In the UV1 range, the sum of PAs multiplied by corresponding coefficients, regardless of which Gaussian is assumed as slit function, is very similar because the spectral structure caused by the slit width change is dominant

**C5.** Fig. 5: The 1st order spectra look wrong. According to Eq. 9, they are 0 in the center of the wavelength window and increase towards the edges (compare Fig. 10 in BE17). The presented spectra look the other way round.

**R5.** The presented spectra for the zero and first order polynomial coefficients are defined as $\frac{\partial lnI}{\partial p} \times \Delta p_0$ and $\frac{\partial lnI}{\partial p} \times \Delta p_1 (\lambda - \bar{\lambda})$). The spectral features by multiplying $(\lambda - \bar{\lambda})$ in PAs are not clearly distinguished in the presented spectra due to the scaling by $\Delta p$. It is clearly shown if $\Delta p$ is taken out as shown in **S4**.

[Figure]

S4. Comparison of PAs for zero and first order polynomial fit.

**C6**. Fig. 9: Specify the time range of the presented data.

**R6.** The time range of the resented data has been specified in the corresponding caption and moreover this figure has been changed to Table 1, according to the reviewer 2's comment.

**C7**. Fig. 10: The unit on the x axis must be DU per km or per vertical layer. Please specify.

**R7.** The ozone in the unit of DU represent the vertically integrated column for the given altitude range (i.e., DU at each vertical layer) and hence the unit on the x axis should be DU.

**C8**. Fig. 10: Abbreviation "MB" is not defined.

**R8.** In the revised manuscript, "MB" and "SD" have been spelled out to Mean Bias and Standard Deviation in the x axis title.

---

## Author Comment (AC2) · 26 May 2019

The author would like to thank Anonymous referee #2 for the constructive and helpful suggestions on this manuscript.

We replied to 3 major comments and 11 technical comments.

**General Comments**

This paper is well organized to describe a methodology for reducing the spectral fit residuals. The subject of the paper is appropriate to AMT. Below are a few comments concerning clarifications / extensions for consideration in the final publication in AMT.

**Major comments**

**C1.** The PROFOZ algorithm applies the pre-estimated, pixel dependent "soft calibration" factors to the normalized radiances, while conducting the ozone profile retrievals. The "soft calibration" factors seem, by design, accounting for the imperfectness of OMI L1B earthshine radiances and solar irradiances calibration, parameterization of the pixel & wavelength dependent ISRFs, and forward model parameters (absorption cross-sections, surface albedo) etc. The PROFOZ also fits scaling factors for the pre-estimated mean spectral fit residuals (Liu 2010 a, b) for UV1 and UV2 bands accordingly, to account for the remaining systematic errors that were not fully removed from "soft calibration" process. This work suggests fit additional ISRF PA coefficients is necessary for OMI ozone profile retrievals. It seems there might some degeneracy among these approaches. The authors should elaborate whether employing pixel & temporal dependent 'soft calibration' factors, or fitting the mean spectral residuals could also achieve the goals same to employing the presented PA approach, in terms of reducing the spectral fit residuals. Are the Jacobians of these PA coefficients, orthogonal to the pre-estimated mean fitting residual spectra, or any other Jacobians of parameters in the retrieval vector?

**R1**. - As this review pointed out, the soft calibration could partly take into account the remaining systematic errors including the spectral structures due to slit function errors, but it should be taken as a last resort after the known physically treatable errors are considered separately. The applied soft spectra were derived from clear-sky tropical measurements in July 2006 and then applied to everywhere and every day. However, the PAs are calculated at each satellite pixel based on the physics associated with slit convolution proposed in Berlie et al. (2017), and iteratively adjusted with the retrieved coefficients. Therefore, the presented PA approach works much better than soft calibration to reduce the fitting residuals and retrieval errors caused by slit function errors.

- Several peaks of soft spectrum are matched with those of PA jacobians, but the soft spectrum is uncorrelated with PAs (with correlation less than 0.1 in UV1 and 0.3 in UV2) because of other dominant factors causing much higher spectral residuals in the soft spectrum (S1). In addition, PA spectra show a weak correlation with other Jacobians within 0.3 for UV1 variables, but for within 0.1 for UV2 variables (S2). In the revised manuscript, this discussion has been added such as "It should be noted that these spectral structures are weakly correlated with the partial derivatives of radiances with respect to other state vectors (ozone, BrO, cloud fraction, surface albedo, radiance/irradiance shift, radiance/ozone cross section shift, Ring/mean fitting residual scaling factor) within $\pm$ 0.3 and $\pm$ 0.1 in the UV 1 and UV 2, respectively."

[Figure]

**S1**. Comparison of soft spectrum with PA jacobians with respect to $\Delta w$ and $\Delta k$, respectively.

**S2**. Correlation of jacobians from the other fitting parameters with PA jacobians with respect to $\Delta w$ and $\Delta k$ for UV1 and UV2, respectively.

**C2.** The authors should obtain time series of retrieved ISRF PA coefficients. Do they show trends similar to Figure 1? At least for Nadir pixel, if not all pixels.

**R2**. Fig. 1 show the time series of slit function parameters derived from solar irradiance measurements. While the PA coefficients show the deviation from those in Figure 1, so they are not expected to show similar trends as shown in S3. In addition, the PA coefficients can vary from spatial to spatial pixel, and vary along the track for the nadir pixel, so it is not as straightforward to obtain the time series. However, this time series also show the larger variation later in OMI mission, especially in the UV1 due to radiometric calibration issues.

[Figure]

**S3. Time-series of PA coefficients for UV1 and UV2, respectively, spatially collocated to 4.3°E, 50.8°N.**

**C3.** A) The authors evaluated the impacts of with/without retrieving PA coefficients on the bias/RMS between retrieved ozone and in-situ ozonesonde measurements (Figure 9). However, the evaluation only made for the period of 2005 to 2008, when OMI instrument was within design lifetime. The authors should also evaluate the performances using the satellite-ozonesonde measurements in other time periods including 2010 and 2012-2013, when the ISRF characteristics were significantly different than the earlier years, as shown in Figure 1. B) The authors should also add some discussions on the possible reasons

causing these sharp changes of ISRF characteristics.

**R3-a**. As well known, there has been concern over the row anomaly effects appearing in 2007 and becoming serious in early 2009, causing trend errors of OMI tropospheric ozone as reported in Huang et al. (2017). Therefore, the period of 2005 to 2008 is focused on the evaluation of including PAs on ozone profile retrievals to avoid any interference with row-anomaly impact. "This evaluation is limited to the period of 2005 through 2008 to avoid interferences with row-anomaly effects appearing in 2007 and becoming serious in early 2009 (Schenkeveld, et al 2017)" has been added in Section 3.2 of the revised manuscript to clarify why the period of 2005 to 2008 is targeted.

**R3-b**. To explain the sharp changes of ISRF characteristics, "The sharp change and random-noise of these derived slit function parameters might be influenced by the decreasing signal-to-noise ratio (SNR) of solar spectra later in the OMI mission and radiometric errors in solar irradiance due to row anomaly (Sun et al., 2017)." has been added in the revised manuscript.

**Technical comments**

**C1.** Have the authors evaluated the impacts of this methodology on the L2 retrieval throughput/yields?
**R1.** There is no significant impact on throughput. The number of successful retrievals for one orbit measurements is 10880 (standard Gaussian, w/o PA), 10880 (super Gaussian, w/o PA), and 10884 (standard Gaussian, with PA), and 10883 (super Gaussian, with PA)

**C2.** Line 29, use the statistical numbers on the bias/RMS differences to replace the word "substantial".
**R2.** The manuscript has been revised to accept this comment as followings.
 - (Abstract) "Comparisons with ozonesondes demonstrate noticeable improvements with the use ofwhen using PAs for both standard and super Gaussians, especially for reducing the systematic biases in the tropics and mid-latitudes (mean biases of tropospheric column ozone reduced from -1.4 ~ 0.7 DU to 0.0 ~0.4 DU) and reducing the standard deviations of tropospheric ozone column differences at high-latitudes (by 1 DU for the super Gaussian)."
  - (Line 329) "clearly shows that including PAs to account for ISRF differences significantly reduces mean biases below 10 km" ➔ "clearly shows that including PAs to account for ISRF differences significantly reduces mean biases ofby up to ~ 5 % below 10 km"
  - (Line 383) "Using super Gaussians, the TCO comparison shows significant improvement in mean biases in mid-latitudes and in standard deviations in high-latitudes. Using standard Gaussians, the TCO comparison also shows significant improvement in mean biases in the tropics" ➔ "In the TCO comparison between OMI and ozonesonde, the mean biases are reduced by 0.2 (0.6) DU and 0.6 (1.4) DU in the tropics (mid-latitude) when super and standard Gaussians are linearized, respectively."

**C3.** Line 47, the authors should consider to revise "by narrow and weak absorption features of the temperature-dependent Huggins bands (320-360 nm)" to "by the 320-330 nm absorption features residing in the temperature-dependent Huggins bands.", since neither this work nor the referenced studies utilized spectral region > 330 nm in the OMI ozone profile retrievals. "narrow and weak" are general terms and might subjective, e.g., this statement will break down. When comparing within the Chappuis bands, the refereed portion of Huggins bands (>320 nm) is no longer weak.
**R3.** According to this comment, the indicated sentence has been revised to "by the 310-330 nm absorption features residing in the temperature-dependent Huggins bands".

**C4.** Line 50, I will suggest to cite the following studies on OMI ozone profile retrievals, since [1] they made use of the ISRFs from Dirksen et al., [2006] cited a few times in this work, [2] the quality

evaluation have been conducted by the comparison with in-situ ozonesonde measurements, [3] same to Liu et al., 2010 cited in this work, these studies were conducted prior to the era of including PA coefficients in the retrieval vector.

**R4.** We appreciate this suggestion. The suggested references have been cited such as "For space-borne instruments, ISRFs are typically characterized as a function of the detector dimensions using a tunable laser source prior to the launch (Dirksen et al., 2006; Liu et al., 2015; van Hees et al., 2018) and directly used in ozone profile retrievals (e.g., Kroon et al., 2011; Mielonen et al., 2015; Fu et al., 2013; 2018)"

**C5.** Line 60, might be a typo (radiance repeated twice)?. Do the authors mean "differences in stray light between radiance and irradiance" or "differences in stray light among OMI measurements"?

**R5.** It is printed-word. It should be "differences in stray light between radiance and irradiance"

**C6.** Line 61, It seems that "intra-orbit instrumental changes" is duplicating the statement of "the instrument temperature change". Please clarify (or remove one).

**R6.** It has been clarified such as "Slit function differences between radiance and irradiance could exist due to scene heterogeneity, differences in stray light between radiance and radiance, and intra-orbit instrumental changes (such as instrument temperature change)."

**C7.** Figures 1, 2, 4, 5, 7, 8, 9 and 10, increase the tick length for improving their visibility.

**R7.** All figures have been revised for better visibility.

**C8.** Figures 5, 6, 8, 9 and 10 captions, state the date/time range of the data presented in the figures. It is not where they are all for 1 July 2006, shown in Figure 4 caption.

**R8.** In the revised manuscript, all captions include the date/time range of the data.

**C9.** Figure 9, create a table and move the statistical values to the table. Having all these numbers on the plots resulted in the plots being too busy to read.

**R9.** The corresponding figure has been changed to Table1.

**C10.** Figure 10 # please spell out the "MB" and "SD" in the x axis title, - space suffice to hold the full name and they were not defined in the caption. # Add two panels to show the differences among data sets, as a function of altitude?

**R10.** The a-axis titles have been changed to Mean Bias and Standard Deviation, respectively.

**C11.** Finally, please keep the 'style' of all figures in a similar fashion. e.g., the panel index of Figure 2 (a), (b) and (c) are inside the plots, while the other figures are outside of the plots. I understand that there is no space for the subtitles outside Figure 2b and 2c, due to the x axis labels. The authors should consider to remove those x axis labels, since all panels could share the one of panel c. Similarly, there are unnecessary axis labels in other figures, e.g., Figures 4, 5, 6, 7, 8, 9, and 10, when some subpanels having an identical scale/range across a row and/or a column, authors should consider remove the unnecessary labels in x or y axis, to help readers easily catch key information presented in the figures.

**R11.** Thanks for this detailed suggestion. All figures have been revised for better visibility.